# Alteration in the Synaptic and Extrasynaptic Organization of AMPA Receptors in the Hippocampus of P301S Tau Transgenic Mice

**DOI:** 10.3390/ijms232113527

**Published:** 2022-11-04

**Authors:** Rocio Alfaro-Ruiz, Carolina Aguado, Alejandro Martín-Belmonte, Ana Esther Moreno-Martínez, Jesús Merchán-Rubira, Félix Hernández, Jesús Ávila, Yugo Fukazawa, Rafael Luján

**Affiliations:** 1Synaptic Structure Laboratory, Instituto de Investigación en Discapacidades Neurológicas (IDINE), Departamento de Ciencias Médicas, Facultad de Medicina, Universidad Castilla-La Mancha, Campus Biosanitario, C/Almansa 14, 02006 Albacete, Spain; 2Pharmacology Unit, Department of Pathology and Experimental Therapeutics, Faculty of Medicine and Health Sciences, Institute of Neurosciences, University of Barcelona, 08907 L’Hospitalet de Llobregat, Spain; 3Neuropharmacology and Pain Group, Neuroscience Program, Institut d’Investigació Biomèdica de Bellvitge, IDIBELL, 08907 L’Hospitalet de Llobregat, Spain; 4Centro de Biología Molecular Severo Ochoa, CSIC-UAM, 28049 Madrid, Spain; 5Centro de Investigación Biomédica en Red Sobre Enfermedades Neurodegenerativas, ISCIII, 28049 Madrid, Spain; 6Division of Brain Structure and Function, Faculty of Medical Science, University of Fukui, Fukui 910-1193, Japan

**Keywords:** Alzheimer’s disease, hippocampus, AMPA receptors, immunohistochemistry, electron microscopy, freeze-fracture, AD mouse model

## Abstract

Tau pathology is a hallmark of Alzheimer’s disease (AD) and other tauopathies, but how pathological tau accumulation alters the glutamate receptor dynamics driving synaptic dysfunction is unclear. Here, we determined the impact of tau pathology on AMPAR expression, density, and subcellular distribution in the hippocampus of P301S mice using immunoblot, histoblot, and quantitative SDS-digested freeze-fracture replica labeling (SDS-FRL). Histoblot and immunoblot showed differential regulation of GluA1 and GluA2 in the hippocampus of P301S mice. The GluA2 subunit was downregulated in the hippocampus at 3 months while both GluA1 and GluA2 subunits were downregulated at 10 months. However, the total amount of GluA1-4 was similar in P301S mice and in age-matched wild-type mice. Using quantitative SDS-FRL, we unraveled the molecular organization of GluA1-4 in various synaptic connections at a high spatial resolution on pyramidal cell spines and interneuron dendrites in the CA1 field of the hippocampus in 10-month-old P301S mice. The labeling density for GluA1-4 in the excitatory synapses established on spines was significantly reduced in P301S mice, compared to age-matched wild-type mice, in the strata radiatum and lacunosum-moleculare but unaltered in the stratum oriens. The density of synaptic GluA1-4 established on interneuron dendrites was significantly reduced in P301S mice in the three strata. The labeling density for GluA1-4 at extrasynaptic sites was significantly reduced in several postsynaptic compartments of CA1 pyramidal cells and interneurons in the three dendritic layers in P301S mice. Our data demonstrate that the progressive accumulation of phospho-tau is associated with alteration of AMPARs on the surface of different neuron types, including synaptic and extrasynaptic membranes, leading to a decline in the trafficking and synaptic transmission, thereby likely contributing to the pathological events taking place in AD.

## 1. Introduction

The postsynaptic responses of neurons to glutamate in the hippocampus are mediated by a range of ionotropic and metabotropic glutamate receptors [1,2]. Ionotropic glutamate receptors mediate the fast component of glutamatergic responses, mostly consisting of AMPA (α-amino-3-hydroxy-5-methyl-4-isoxazole propionic acid) and NMDA (*N*-methyl-*D*-aspartate) receptors [3]. The fast kinetics of AMPA receptors (AMPARs) differentiate them from NMDARs, allowing a rapid depolarization of the postsynaptic membrane and propagation of electric signals between neuronal cells [3]. These receptors also play fundamental roles in synaptic plasticity, the underlying molecular mechanism of learning and memory [4]. Consequently, disruption of the signaling mediated via AMPARs is implicated in Alzheimer’s disease (AD), thereby making them interesting therapeutic targets for neurological disorders [5]. AD is commonly associated with elevated levels of amyloid-beta (Aβ) peptide and hyperphosphorylated tau and a decreased number of synapses [6]. Among those pathological hallmarks, accumulating evidence suggests that synaptic dysfunction is a major contributor early in disease pathogenesis prior to neuronal loss, which correlates with the loss of memory function and learning ability in AD patients [7].

AMPARs consist of four homologous pore-forming subunits (GluA1-4) encoded by distinct genes: *GRIA1*–*GRIA4* [8]. Furthermore, the molecular diversity of AMPARs is further increased by alternative splicing and post-transcriptional RNA editing [3]. The kinetic properties of AMPARs are determined by the subunit composition and RNA splicing [3]. Out of the four subunits, GluA2 confers unique characteristic to AMPARs determining a role in the rectification properties and Ca^2+^ impermeability of the functional receptors [9]. In the hippocampus, most AMPARs are hetero-oligomers composed of glutamate receptor subunits GluA1/2 and GluA2/3 [10,11,12].

AMPARs are concentrated at the postsynaptic membrane of excitatory synapses in the hippocampus [13,14,15,16]. The content of these synapses is modulated by changes in the number of AMPARs, their subunit composition, and post-translational modifications, thus allowing the tight control of synaptic strength required for synaptic plasticity [3,17,18]. Therefore, any alteration in the number and density of AMPARs could contribute to the synaptic and memory deficits that are associated with AD. There is substantial evidence showing a downregulation of AMPARs in AD that seems to be due to Aβ_1–42_-induced internalization and tau-mediated alterations in AMPARs in a variety of transgenic tau mouse models [19].

Using an experimental model that reproduces the Aβ neuropathological hallmark in AD and the same methodological approaches, we previously reported a significant reduction in AMPAR density at excitatory synapses in APP/PS1 mice [20]. However, the role played by tau in AMPAR expression and localization, and whether they could be selectively disrupted at specific postsynaptic sites or neuron types remains unclear. Therefore, in the present study, we used P301S mice, a commonly used tauopathy model with several AD-relevant features [21], to investigate whether hyperphosphorylated tau is associated with changes in the expression and synaptic and extrasynaptic localization of AMPARs in the hippocampus. Here, we show convincing evidence for a reduction in both synaptic and extrasynaptic AMPARs in pyramidal cells and interneurons of the hippocampal CA1 field in P301S mice.

## 2. Results

### 2.1. Age-Dependent Alteration in the GluA1-4 Expression in the Brain of P301S Mice

The alteration in the AMPAR expression in the brain of P301S and age-matched wild-type mice was determined using a GluA1-4-specific antibody in conventional histoblots [22] at 3 and 10 months of age (Figure 1A–F). In wild-type mice, immunolabeling for GluA1-4 was widely distributed in the brain at the two ages, showing marked region-specific differences, with the strongest labeling in the hippocampal formation, superficial layers of the neocortex, the molecular layer of the cerebellum, the pre-subiculum, and the septum (Figure 1A,C,D,F). Moderate labeling was found in deeper layers of the neocortex and caudate putamen, and weak labeling was detected in the thalamus and midbrain nuclei (Figure 1A,C,D,F). This GluA1-4 expression pattern was very similar in P301S mice at 3 months (Figure 1B,C), but quantitative analyses showed a significant decrease in labeling in the hippocampus proper and pre-subiculum of the hippocampal formation at 10 months of age (Figure 1E,F).

Using the same methodological approach, we next focused on the hippocampus and explored the laminar expression pattern of AMPARs (Figure 1G–L). GluA1-4 was widely expressed in all hippocampal subfields at the two ages and genotypes (Figure 1G–L). In the CA1 and CA3 regions of wild-type mice, the stratum radiatum (SR) showed the highest GluA1-4 expression levels, followed by the strata oriens (SO) and lacunosum-moleculare (SLM) (Figure 1G,I,J,L). The stratum lucidum in the CA3 region showed a weak expression level (Figure 1G,I,J,L). In the dentate gyrus, the expression of GluA1-4 was weak in the hilus and moderate in the molecular layer (Figure 1G,I,J,L). The stratum pyramidale of the CA1 and CA3 regions and the granule cell layer of the dentate gyrus showed the weakest GluA1-4 staining. These labeling patterns of GluA1-4 in wild-type mice were similar to those observed in P301S mice at the two ages (Figure 1H,I,K,L). Quantitative analyses of the immunoreactivities confirmed that the expression levels of GluA1-4 in all subfields and dendritic layers analyzed were unchanged in both wild-type and P301S mice at 3 months of age (Figure 1I). However, the expression of GluA1-4 was significantly reduced in the molecular layer and hilus of the dentate gyrus of P301S mice compared to age-matched wild-type controls mice at 10 months of age (Figure 1L).

### 2.2. Early and Differential Alteration of GluA1 and GluA2 Brain Expression in P301S Mice

Next, we explored the expression of GluA1 and GluA2, the two most common AMPAR subunits in the hippocampus [23], using the histoblot technique at 3 and 10 months of age (Figure 2). In wild-type mice, the expression of GluA1 was strongest in the hippocampal formation, followed by the molecular layer of the cerebellum, the pre-subiculum, and the septum, and weak in the remaining regions at both 3 and 10 months (Figure 2A,C,D,F). In P301S mice, the expression of GluA1 at 3 months was mostly unaltered (Figure 2B,C), although at 10 months, it was significantly reduced in the hippocampus and pre-subiculum of the hippocampal formation (Figure 2E,F).

The expression of GluA2 in wild-type mice was strongest in the hippocampal formation, presubiculum, and septum, followed by the cortex, molecular layer of the cerebellum, and the caudate putamen, and weak labeling was detected in the thalamus and midbrain nuclei (Figure 2G,I,J,L). This GluA2 expression pattern was significantly reduced in all brain regions analyzed in P301S mice at 3 months (Figure 2H,I) and only decreased labeling was detected in the hippocampus and pre-subiculum at 10 months of age (Figure 2K,L).

In the hippocampus, the regional and laminar expression of GluA1 and GluA2 was similar (Figure 3). In the CA1 and CA3 regions, the highest expression of GluA1 and GluA2 in wild-type mice was strongest in SR, followed by SLM and SO, and in the dentate gyrus, expression was strong in the hilus and moderate in the molecular layer (Figure 3). These labeling patterns changed in P301S mice in a subunit-, laminar-, and age-dependent manner. Quantitative analyses of the immunoreactivities showed that the expression of GluA1 at 3 months was increased in SO and SR of the CA1 field, SO of the CA3 field, and the molecular layer of the dentate gyrus (Figure 3C). However, GluA1 expression at 10 months was significantly reduced in SO, SR, and SLM of the CA1 field; SLM of the CA3 field; and the molecular layer and hilus of the dentate gyrus (Figure 3C). Differently to GluA1, the expression of GluA2 was significantly decreased in all subfield and lamina of the hippocampus at 3 months of age (Figure 3I) and only in SLM of the CA1 and CA3 fields and hilus of the dentate gyrus (Figure 3L).

### 2.3. Reduction in the Expression of AMPA Receptors in the Hippocampus of P301S Mice

We further evaluated the expression of GluA1, GluA2, and GluA1-4 in the hippocampus at 3 and 10 months of age using immunoblots of membrane fractions (Figure 4), which reaffirmed the data obtained using histoblots. The three antibodies revealed a single immunoreactive band with an estimated molecular mass of 100 kDa (Figure 4A). The levels of GluA2 protein were significantly reduced at 3 months but unchanged for GluA1 and GluA1-4 in P301S mice (Figure 4A,B). At 10 months, the levels of GluA1 and GluA2 proteins were significantly reduced in P301S mice, with a tendency of a reduction for GluA1-4 (Figure 4A,C). Overall, these results indicate detectable changes in the hippocampal expression of key AMPAR subunit proteins in P301S mice.

### 2.4. Differential Reduction of AMPAR s in the Excitatory Synapses in the Spines of P301S Mice

Using the SDS-FRL technique, we analyzed the distribution and densities of GluA1-4 in the excitatory synapses established on pyramidal cell spines and on interneuron dendrites in the strata oriens (SO), radiatum (SR), and lacunosum-moleculare (SLM) of the CA1 field. Replicas were obtained from 10-month-old wild-type and P301S mice, reacted with the rabbit anti-GluA1-4 antibody and only labeled clusters of IMPs on the E-face, representing the postsynaptic membrane specialization of glutamatergic synapses, were analyzed.

First, we performed the analysis at the excitatory synapses in the CA1 pyramidal cell spines of wild-type and P301S mice (Figure 5A–F). In wild-type mice, immunoparticles for AMPARs were distributed over the entire postsynaptic membrane specializations of spines with no apparent clustering in SO, SR, and SLM (Figure 5A–C). In contrast, fewer AMPAR immunoparticles were detected on the excitatory synapses of spines in the three CA1 dendritic layers in P301S mice (Figure 5D–F). The possible changes in the density of synaptic AMPARs between the wild-type and P301S mice were tested, revealing that although the density of labeling varied between synapses (Figure 5G,H; Table 1), there was a significant reduction in the GluA1-4 density in the excitatory synapses on the spines in SR and SLM in P301S mice (SR: 377.41 ± 46.73 immunoparticles/μm^2^, n = 52 synapses; SLM: 375.59 ± 47.96 immunoparticles/μm^2^, n = 33 synapses) compared to age-matched wild-type mice (SR: 692.70 ± 44.43 immunoparticles/μm^2^, n = 48 synapses; SLM: 833.86 ± 38.45 immunoparticles/μm^2^, n = 51 synapses) (unpaired *t*-test, **** *p* < 0.0001) (Figure 5H; Table 1). However, the density was unaltered in the SO (wild type: 373.25 ± 49.93 immunoparticles/μm^2^, n = 31 synapses; P301S: 245.44 ± 26.64 immunoparticles/μm^2^, n = 35 synapses) (unpaired *t*-test, *p* = 0.56) (Figure 5H; Table 1).

The size of the postsynaptic membrane specialization on spines throughout the CA1 field was not significantly different between wild-type and P301S mice (unpaired *t*-test, *p* > 0.05) (Table 1). The number of immunoparticles for GluA1-4 in those excitatory synapses was highly variable, although a linear correlation between the number of immunoparticles and the area of synaptic sites was found for both wild-type and P301S mice, except in SR in P301S mice (Appendix A; Table 1), indicating a similar density of AMPARs between most excitatory synapses in normal and pathological conditions.

### 2.5. Reduction of AMPARs in the Excitatory Synapses in the Interneurons of P301S Mice

Next, we investigated whether synaptic AMPARs are also altered in the excitatory synapses on interneurons in SO, SR, and SLM of the CA1 field using the SDS-FRL technique in both wild-type and P301S 10-month-old mice (Figure 6A–F). Immunoparticles for GluA1-4 in interneurons were observed over the postsynaptic membrane specializations in wild-type mice (Figure 6A–C) but were consistently fewer in P301S mice (Figure 6D–F). The possible changes in the density of synaptic GluA1-4 were quantified (Figure 6G; Table 1). This analysis revealed a significant reduction in the GluA1-4 density in the excitatory synapses on the dendritic shafts in the three dendritic layers in P301S mice (SO: 230.02 ± 19.28 immunoparticles/μm^2^, n = 78 synapses; SR: 300.01 ± 33.29 immunoparticles/μm^2^, n = 56 synapses; SLM: 548.08 ± 49.05 immunoparticles/μm^2^, n = 33 synapses) compared to age-matched wild-type mice (SO: 367.01 ± 39.31 immunoparticles/μm^2^, n = 45 synapses; SR: 764.07 ± 38.37 immunoparticles/μm^2^, n = 61 synapses; SLM: 967.70 ± 43.95 immunoparticles/μm^2^, n = 47 synapses) (Mann–Whitney *U* test ** *p* < 0.01, **** *p* < 0.0001) (Figure 6H; Table 1).

The size of the excitatory synapses established on interneuron dendrites showed no significant differences between wild-type and P301S mice (unpaired *t*-test, *p* > 0.05) (Table 1). The number of GluA1-4 immunoparticles in those excitatory synapses was highly variable in both wild-type and P301S mice, but a linear correlation between the number of immunoparticles and area of synapses was found (Appendix A; Table 1), indicating that the AMPAR density was similar between the excitatory synapses established on interneuron dendrites in normal and pathological conditions.

### 2.6. Reduction of Extrasynaptic AMPARs in the Hippocampus of P301S Mice

Extrasynaptic AMPARs are widely expressed in the hippocampus. To explore whether they also undergo changes in pathological conditions, we analyzed the distribution of GluA1-4 in the extrasynaptic membranes of pyramidal cells and interneurons throughout the CA1 field using the same methodological approach as for synaptic AMPARs (Figure 7A–F). Extrasynaptic GluA1-4 was detected along the surface of the pyramidal cell spines and shafts and interneuron dendrites associated with isolated IMPs or small clusters of IMPS in both wild-type and P301S mice (Figure 7A–F). The quantitative analysis in the spines and shafts of pyramidal cells and interneuron dendrites throughout SO, SR, and SLM showed a significant decrease in extrasynaptic GluA1-4 in the spines of SR and SLM, in the dendritic shafts of pyramidal cells in SO, and in the dendrites of interneurons in SR of P301S mice compared to age-matched wild-type mice (Mann–Whitney *U* test, ** *p* < 0.01, **** *p* < 0.0001, ***** *p* < 0.0001) (Figure 7G–I, Table 2). No differences between wild-type and P301S mice in other dendritic compartments were observed (Mann–Whitney *U* test, *p* > 0.05) (Figure 7G–I, Table 2). In summary, parallel to the reduction for synaptic AMPARs, these results indicate that extrasynaptic AMPARs are also reduced in P301S mice at 10 months of age.

## 3. Discussion

Synaptic dysfunction is an early event in the pathophysiology of AD and is most closely correlated with cognitive deficits in AD patients [24,25,26]. However, the molecules that are affected at identified synapses are not yet fully known. Moreover, such information is considered critical to allow for the development of more effective therapeutic treatments for AD. Increasing evidence indicates that the loss of synapses correlates better with tau pathology than with amyloid pathology [27,28,29,30]. Thus, we focused on tauopathy and the possible changes in the molecular organization of AMPARs as key players in the modulation of excitatory synaptic transmission and synaptic plasticity. In this work, we identified GluA1 and GluA2 as AMPARs subunits that are altered and differentially regulated in the brain in the early stages of tauopathy and further demonstrated the AMPAR organization in various synaptic connections at high spatial resolution and detection sensitivity in normal and pathological conditions using quantitative SDS-FRL. Our data suggest that tauopathy impacts synaptic and extrasynaptic receptors, leading to a decline in the trafficking and synaptic transmission of AMPARs in the excitatory synapses on pyramidal cell spines and interneuron dendrites in the CA1 field of the hippocampus in P301S mice. These hippocampal circuit alterations may contribute to the failure in network activities across the brain and thereby underly the deficits described in this animal model of AD.

### 3.1. Differential Expression of AMPAR Subunits in the Hippocampus of P301S Mice

AMPARs are widely expressed in the brain, and the hippocampus is among the regions with the highest expression for the four subunits [10]. In the present study, we showed by histoblot that GluA1, GluA2, and GluA1-4 were widely expressed in the hippocampus at 3 and 10 months and the labeling was particularly strong in the dendritic layers of the CA1 field, where excitatory synapses are established in spines and interneuron dendrites. The histoblot patterns obtained agree well with previous autoradiographic studies using AMPA receptor-selective radioligands [31] and are consistent with the cellular distribution of different AMPAR subunit mRNAs and proteins [8,10,32,33,34]. More importantly, in tau pathology, we revealed that the most prominent changes take place within the hippocampus proper and presubiculum of the hippocampal formation, with a significant decrease in the protein expression of GluA1 and GluA2 in 10-month-old P301S mice. The unaltered protein expression of GluA1-4 is consistent with previous findings in APP/PS1 mice and is very likely due to internalization of AMPARs [20]. Alternatively, this could be due to compensatory mechanisms at excitatory synapses. Overall, our data are consistent with the alteration of AMPARs described in rTg4510 mice [35], tau-transfected neurons [36], and neuronal cultures overexpressing P301L tau [37]. Our findings are also consistent with recent Western blotting approaches describing a significant decrease in the protein expression of GluA1 and GluA2 in the hippocampus of AD patients [20] and with previous autoradiographic studies indicating that the binding of glutamate to AMPARs was significantly decreased, particularly in the CA1 region [38,39,40].

Another important feature of AMPARs concerns their subunit composition, which has been proposed to determine their mode of trafficking [41]. Synaptic and surface AMPARs in the hippocampus are mainly composed of GluA1/2 and GluA2/3 heterodimers, with GluA1/2 being the most common [23,42]. An interesting finding of the present study is the differential regulation of GluA1 and GluA2 in the brain of P301S mice at 3 months, an age at which hippocampal synapse loss and impaired synaptic function are detected before fibrillary tau tangles emerge [21]. We found a dramatic downregulation in the expression of GluA2 in the brain and all hippocampal regions analyzed, parallel with an upregulation in the expression of GluA1 in a brain-region- and laminar-dependent manner. The presence of GluA2 is of functional importance because this subunit is AMPARs Ca^2+^ impermeable [43]. GluA2-containing AMPARs are more stable at the synapse due to the interaction of GluA2 with synaptic proteins that promote the retention of the receptors in the plasma membrane [11,44,45,46]. In addition, the GluA2 subunit seems to stabilize dendritic spines in the hippocampus [47,48]. In line with these properties, by demonstrating a dramatic reduction in GluA2 expression at 3 months of age, likely compensated with an upregulation of GluA1 in some dendritic layers, our data favors a possible role of GluA1/2 receptor subunit-specific trafficking in tau pathology driving the decline of AMPARs at excitatory synapses that could culminate in synapse and spine loss in P301S mice [21].

### 3.2. Reduction of Synaptic AMPARs in Pyramidal Cells and Interneurons in P301S Mice

Hippocampal microcircuits in the CA1 field include excitatory principal cells and different classes of GABAergic interneurons, which exert inhibitory control of different compartments of principal cells and other interneurons [49,50]. In situ hybridization and immunohistochemical studies have shown that AMPAR subunit (GluA1-4) mRNAs and proteins are expressed in both pyramidal cells and interneurons in the CA1 field of the hippocampus [8,10,32,33,34], where they are fundamental in defining their functional properties. In the two neuron populations, AMPARs are enriched in most excitatory synapses, where they mostly reside in the postsynaptic membrane opposite the presynaptic active zone and along the extrasynaptic membrane [14,15,16,51]. Using an antibody against the highly conserved extracellular amino acid residues of the AMPAR subunits GluA1-4 and the highly sensitive SDS-FRL method, with nearly one gold particle-one functional channel sensitivity [52], we were able to unravel the nanoscale organization of AMPARs in various synaptic connections in the CA1 field in physiological conditions and in tauopathy. Our direct measurements revealed an uneven distribution of GluA1-4 immunoparticles within individual IMP clusters in excitatory synapses and a great variability in the AMPAR content at individual synapses from neuron to neuron and from spine to spine, a finding consistent with previous reports using post-embedding immunogold techniques [13,14,15,16]. In agreement with these studies, we confirmed that the AMPAR content of the excitatory synapses on the spines and interneurons correlates with the synaptic area in normal and pathological conditions. These relationships applied to the three dendritic layers both in wild-type and P301S mice and are similar to those recently described in the APP/PS1 model of AD [20].

The localization and number of AMPARs at the cell surface membrane are dynamically regulated and critical for LTP and LTD [53]. The animal model used in the present study, the P301S mice, exhibits deficits in synaptic function, including impaired LTP in the CA1 field [21], and increasing evidence indicates that the accumulation of abnormal tau is synaptotoxic [37,54,55]. The exact molecular mechanisms by which tauopathy leads to synaptic dysfunction and loss remain unclear. The results obtained in this study demonstrated a significant reduction in the synaptic localization of AMPARs at the excitatory synapses on the spines of CA1 pyramidal cells in the stratum radiatum and lacunosum-moleculare but not in the stratum oriens, indicating laminar-specific differences in response to tau accumulation. Our data are consistent with previous reports suggesting that hyperphosphorylated tau reduces the trafficking of the glutamate receptor subunits GluA1 and GluA2/3 to PSD-95 [37].

Synaptic disturbances in the excitatory to inhibitory balance in hippocampal circuits contribute to the progression of AD [56]. In addition to the multiple studies describing dysfunctions in excitatory neuronal functions, impaired inhibition by interneurons has also been associated with cognitive deficits in AD [56,57,58]. Our results demonstrate that, on average, excitatory synapses contained more GluA1-4 immunoparticles than the most labeled excitatory synapses on the spines, consistent with previous studies [13,20]. In addition, an interesting finding of the present study is the significant reduction in synaptic GluA1-4 in interneuron dendrites in the three dendritic layers of the CA1 field. Given the role of AMPARs in interneurons at the CA1 neuronal network level contributing to the generation and/or maintenance of rhythmic oscillatory activities [49,50], our data suggest an impairment of such a role in the hippocampus in tauopathy.

### 3.3. Reduction of Extrasynaptic AMPARs in P301S Mice

AMPARs are dynamic components of the excitatory hippocampal synapses showing lateral mobility between the synaptic and extrasynaptic membranes, where the AMPAR subunit complex can be internalized through endocytosis [59,60] and where they contribute to excitatory transmission by controlling the synaptic efficacy [61,62]. Previous immunoelectron microscopy showed a high proportion of extrasynaptic AMPAR in hippocampal neurons [13,34,63]. Consistent with these studies, using quantitative SDS-FRL techniques, we demonstrated a high proportion of GluA1-4 at extrasynaptic sites, including the dendritic shafts and spines of CA1 pyramidal cells and dendrites of interneurons. Such a large amount of extrasynaptic AMPARs could serve as a reserve pool that can rapidly move in and out of synaptic membranes [61]. More importantly, an interesting finding of the present study is that the labeling density for GluA1-4 at extrasynaptic sites was significantly decreased in several compartments of the pyramidal cells and interneurons in P301S mice. Given that the availability of an extrasynaptic pool can influence how rapidly AMPARs can be inserted into synaptic sites after stimulation [61], our data suggest that tauopathy-associated defects in synaptic plasticity [21] can arise from defective AMPAR trafficking.

Recent studies demonstrated that tau is required for the function of extrasynaptic NMDARs [64] and that they are mostly increased in P301S mice [65]. In this study, we identified a reduction in both synaptic and extrasynaptic AMPARs in tau pathology in the two main neuronal populations of the CA1 field: pyramidal cells and interneurons. Our data demonstrate that molecular changes at the synapse and extrasynaptic membranes can be studied with an extremely high spatial resolution. Alteration in the surface exchange of AMPAR between synaptic and extrasynaptic underlies the functional synaptic changes that occur in Alzheimer’s disease models [66]. Thus, the inhibition of Ca^2+^/calmodulin-dependent protein kinase II (CaMKII), a molecule playing a key role in AMPAR trafficking and synaptic plasticity, induced by treatment with oligomeric Aβ leads to dendritic spine loss via the destabilization of surface AMPARs [66]. Here, our quantitative ultrastructural data support the view that tau might induce synaptic dysfunction [36] through the decline in surface AMPARs.

In summary, the present work provides evidence that hippocampal GluA1 and GluA2 subunits are altered in tau pathology. This is the first detailed description of the altered molecular organization of AMPARs in the hippocampus in P301S mice, presenting quantitative evidence that a reduction in GluA1-4 takes place both in the synaptic and extrasynaptic compartments of CA1 pyramidal cells and interneurons. Given the link between the GluA1 and GluA2 subunits and tau pathology, our data provide novel insights into the role of AMPARS in mediating synaptic dysfunctions and offer avenues for the development of novel therapeutics for AD.

## 4. Material and Methods

### 4.1. Animals

We used transgenic mice P301S for the human Tau gene and wild-type control littermates. The P301S mouse model, obtained from Jackson laboratory (B6;C3-Tg(Prnp-MAPT*P301S)PS19Vle/J), carries a mutant (P301S) human MAPT gene encoding T34-tau isoform (1N4R) driven by the mouse prion-protein promoter (Prnp) on a C57BL/6J genetic background. For analysis, we selected animals at 3 months of age, characterized by no sign of pathology [21], and 10 months of age, characterized by widespread neurofibrillary tangles accumulation, impaired memory, spatial learning and LTP, impaired synaptic function, and neuronal loss [21]. A total of 12 mice aged 3 months (n = 3 for WT, n = 3 for P301S used for histoblotting; n = 3 for WT, n = 3 for P301S used for immunoblotting) and 18 mice aged 10 months (n = 3 for WT, n = 3 for P301S used for histoblotting; n = 3 for WT, n = 3 for P301S used for immunoblotting; and n = 3 for WT, n = 3 for P301S used for immunoelectron microscopy) were analyzed. All mice were housed at the Centro de Biología Molecular Severo Ochoa animal facility. Mice were housed four per cage with food and water available ad libitum and maintained in a temperature-controlled environment on a 12/12 h light-dark cycle with light onset at 07:00 h. Animal housing and maintenance protocols followed the guidelines of Council of Europe Convention ETS123, recently revised as indicated in Directive 86/609/EEC. Animal experiments were performed under protocols (P15/P16/P18/P22) approved by the Institutional Animal Care and Utilization Committee (Comité de Ética de Experimentación Animal del CBM, CEEA-CBM, Madrid, Spain).

For immunoblotting and histoblotting, animals were deeply anesthetized by intraperitoneal injection of ketamine/xylazine 1:1 (ketamine, 100 mg/kg; xylazine, 10 mg/kg), the hippocampus was dissected, frozen rapidly in liquid nitrogen, and stored at −80 °C. For immunohistochemistry experiments at the electron microscopic level, using the pre-embedding immunogold technique, animals were firstly deeply anaesthetized by intraperitoneal injection of ketamine-xylazine 1:1 (ketamine, 100 mg/kg; xylazine, 10 mg/kg) and then transcardially perfused with ice-cold fixative containing 4% (*w*/*v*) paraformaldehyde with 0.05% (*v*/*v*) glutaraldehyde in 0.1 M phosphate buffer (PB, pH 7.4) for 15 min. After perfusion, brains were removed from the skull, and tissue blocks were washed thoroughly in 0.1 M PB. Coronal 60-μm-thick sections were cut on a Vibratome (Leica V1000). For the SDS-FRL experiments, animals were anesthetized with sodium pentobarbital (50 mg/kg, i.p.) and perfused transcardially with 25 mM PBS for 1 min, followed by perfusion with 2% paraformaldehyde in 0.1 M phosphate buffer (PB) for 12 min. After perfusion, brains were removed, and the hippocampi were dissected and cut into sagittal slices (130 µm) using a Microslicer (Dosaka, Kyoto, Japan) in 0.1 M PB.

### 4.2. Antibodies and Chemicals

For SDS-FRL, we used a rabbit anti-GluA1-4 (pan-AMPA) receptor polyclonal antibody (D160), whose preparation, purification, and characterization have been described recently [20]. For histoblot, we used a guinea pig pan-GluA1-4 receptor antibody (GP-Af580; aa. 717–754 of mouse AMPAR; Frontier Institute Co., Japan), characterized previously [67]; a rabbit pan-GluA1 subunit antibody (Rb-Af690; aa. 841–907 of mouse AMPAR; Frontier Institute Co., Sapporo, Japan); and a rabbit pan-GluA2 subunit antibody (RB-Af1050-Af580; aa. 847–863 of mouse AMPAR; Frontier Institute Co., Sapporo, Japan). For the pre-embedding immunogold technique, we used a polyclonal rabbit antibody anti-GluA2/3 (AB1506; Chemicon, Temecula, CA, USA). These antibodies were raised against synthetic peptides derived from intracellular C-terminal sequences of the GluA2 and GluA3 subunits [32]. For double-SDS-FRL, we used a mouse monoclonal antibody against the GluN1 subunit of the NMDA receptor (MAB363, Millipore Bioscience Research Reagents, Burlington, MA, USA). The characteristics and specificity of GluN1 were characterized previously [63,68,69,70]. While the guinea pig GluA1-4 antibody, the rabbit anti-GluA1, and anti-GluA2 antibodies worked well for histoblots, they produced relatively weak labeling in SDS-FRL. Therefore, we used rabbit anti-GluA1-4 antibodies for SDS-FRL and could not perform ultrastructural analyses with the subunit-specific antibodies.

The following secondary antibodies were used: goat anti-mouse IgG-horseradish peroxidase (1:2000; Santa Cruz Biotechnology, Santa Cruz, CA, USA), goat anti-rabbit IgG-horseradish peroxidase (1:15,000; Pierce, Rockford, IL, USA), alkaline phosphatase (AP)-goat anti-mouse IgG (H + L) and AP-goat anti-rabbit IgG (H + L) (1:5000; Invitrogen, Paisley, UK), anti-rabbit IgG conjugated to 5 nm gold particles, and anti-mouse IgG conjugated to 10 nm gold particles (1:100; British Biocell International, Cardiff, UK).

### 4.3. Immunoblots

Hippocampi were homogenized in 50 mM Tris Base, pH 7.4, and Protease Inhibitor Cocktail (Thermo Scientific, Pierce, Rockford, IL, USA) with a pestle motor (Sigma-Aldrich, San Luis, MI, USA). The homogenized tissue was centrifuged for 10 min at 1000× *g* at 4 °C, and the supernatant was centrifuged for 30 min at 12,000× *g* (Centrifuge 5415R, Eppendorf, Hamburg, Germany) at 4 °C, and the pellets containing the membrane extracts were resuspended in 50 mM Tris Base, pH 7.4, and Protease Inhibitor Cocktail (Thermo Scientific, Pierce, Rockford, IL, USA). The protein content of each membrane extract was determined by a BCA Protein Assay Kit (Thermo Scientific, Pierce, Rockford, IL, USA). In total, 25 μg of membrane protein was loaded in sodium dodecyl sulfate polyacrylamide gel electrophoresis (SDS/PAGE) using 7.5% polyacrylamide with loading sample buffer (0.05 M Tris pH 6.8, 2% SDS, 10% glycerol, 0.05% *β*-mercaptoethanol, and 0.001% bromophenol blue). The proteins were transferred to PVDF membranes using a semidry transfer system and immunoblotted with anti-GluA1-4 (1:1000), anti-GluA1 (1:500), anti-GluA2 (1:500), and anti-α-Tubulin (1:500). Protein bands were visualized after the application of a mouse IgG kappa-binding protein coupled to horseradish peroxidase (1:2000) using the enhanced chemiluminescence (ECL) blotting detection kit (SuperSignal West Dura Extended Duration Substrate, Pierce, Rockford, IL, USA). Blots were captured and quantified by densitometry using an LAS4000 MINI (Fujifilm, Tokyo, Japan). A series of primary and secondary antibody dilutions and incubation times were used to optimize the experimental conditions for the linear sensitivity range, confirming that our labeling was well below saturation levels.

### 4.4. Histoblotting

The regional distribution of GluA1, GluA2, and GluA1-4 was analyzed in mouse brains using the histoblot technique [22]. Briefly, horizontal cryostat sections (10 µm) from the mouse brain were overlapped with nitrocellulose membranes moistened with 48 mM Tris-base, 39 mM glycine, 2% (*w*/*v*) sodium dodecyl sulfate, and 20% (*v*/*v*) methanol for 15 min at room temperature (~20 °C). After blocking in 5% (*w*/*v*) non-fat dry milk in phosphate-buffered saline for 1 h, nitrocellulose membranes were treated with Dnase I (5 U/mL), washed, and incubated in 2% (*w*/*v*) sodium dodecyl sulfate and 100 mM β-mercaptoethanol in 100 mM Tris–HCl (pH 7.0) for 60 min at 45 °C to remove adhering tissue residues. After extensive washing, the blots were reacted with affinity-purified anti-GluA1-4 antibodies (0.5 mg/mL), anti-GluA1 antibodies (0.5 mg/mL) or anti-GluA2 antibodies (0.5 mg/mL), in blocking solution overnight at 4 °C. The bound primary antibodies were detected with alkaline phosphatase-conjugated anti-mouse IgG secondary antibodies [22]. A series of primary and secondary antibody dilutions and incubation times were used to optimize the experimental conditions for the linear sensitivity range of the alkaline phosphatase reactions. To compare the expression levels of GluA1, GluA2, and GluA1-4 between the two genotypes (wild type and P301S) and ages (3 and 10 months), all nitrocellulose membranes were processed in parallel, and the same incubation time for each reagent was used for the antibody. Digital images were acquired by scanning the nitrocellulose membranes using a desktop scanner (HP Scanjet 8300). Image analysis and processing were performed using the Adobe Photoshop software (Adobe Systems, San Jose, CA, USA) as described previously [20].

### 4.5. SDS-Digested Freeze-Fracture Replica Labeling

Immunohistochemical reactions at the electron microscopic level were carried out using the SDS-FRL methods as described earlier [52,71]. Briefly, animals were anesthetized with sodium pentobarbital (50 mg/kg, i.p.) and perfused transcardially with 25 mM PBS for 1 min, followed by perfusion with 2% (*w*/*v*) paraformaldehyde in 0.1 M phosphate buffer (PB) for 12 min. The hippocampi were dissected and cut into coronal slices (130 µm) using a Microslicer (Dosaka, Kyoto, Japan) in 0.1 M PB. Next, hippocampal slices containing the CA1 region were trimmed out of the slices and immersed in graded glycerol of 10–30% (*v*/*v*) in 0.1 M PB at 4 °C overnight. Slices were frozen using a high-pressure freezing machine (HPM010, BAL-TEC, Balzers). Slices were then fractured into two parts at −120 °C and replicated by carbon deposition (5 nm thick), platinum (60° unidirectional from horizontal level, 2 nm), and carbon (15–20 nm) in a freeze-fracture replica machine (BAF060, BAL-TEC, Balzers). Replicas were transferred to 2.5% (*w*/*v*) SDS and 20% (*w*/*v*) sucrose in 15 mM Tris-Cl buffer (pH 8.3) for 18 h at 80 °C with shaking to dissolve tissue debris. The replicas were washed three times in 50 mM Tris-buffered saline (TBS, pH 7.4), containing 0.05% (*w*/*v*) bovine serum albumin (BSA), and then blocked with 5% (*w*/*v*) BSA in the washing buffer for 1 h at room temperature. Next, the replicas were washed and reacted with a pan GluA1-4 antibody raised in rabbit (7.3 μg/mL), followed by a mouse monoclonal antibody against the GluN1 subunit of the NMDA receptor (10 μg/mL; Millipore Bioscience Research Reagents), diluted in 25 mM TBS containing 1% (*w*/*v*) BSA overnight at 15 °C. Following three washes in 0.05% BSA in TBS and blocking in 5% (*w*/*v*) BSA/TBS, replicas were incubated in goat anti-rabbit (for pan-GluA1-4) IgGs coupled to 5 nm gold particles (1:30; British BioCell Research Laboratories) diluted in 25 mM TBS containing 5% (*w*/*v*) BSA overnight at room temperature, followed by goat anti-mouse (for GluN1) IgGs coupled to 10 nm gold particles (1:30; British BioCell Research Laboratories) diluted in 25 mM TBS containing 5% (*w*/*v*) BSA overnight at room temperature. When the primary antibody was omitted, no immunoreactivity was observed. After immunogold labeling, the replicas were immediately rinsed three times with 0.05% BSA in TBS, washed twice with distilled water, and picked up onto grids coated with pioloform (Agar Scientific, Stansted, Essex, UK) and examined with an electron microscope (Hitachi H-7650) equipped with a digital camera (Quemesa, EM SIS).

### 4.6. Quantification and Analysis of SDS-FRL Data

The labeled replicas were examined using a transmission electron microscope (JEOL-1400Flash) equipped with a digital high-sensitivity sCMOS camera, and images were captured at magnifications of 30,000×. All antibodies used in this study were visualized by immunoparticles on the exoplasmic face (E-face), consistent with the extracellular location of their epitopes. Non-specific background labeling for anti-GluA1-4 was estimated by counting immunogold particles on the protoplasmic face (P-face) surfaces in wild-type mice. Digitized images were then modified for brightness and contrast using Adobe PhotoShop CS5 (Mountain View, CA, USA) to optimize them for quantitative analysis.

Number and density of AMPAR immunoparticles at synaptic and extrasynaptic sites. Given that not all excitatory synapses in the hippocampus contain AMPARs [13], we analyzed excitatory synapses containing NMDARs as a synaptic marker in double-labeling replicas for both receptors. We determined the number of GluA1-4 immunoparticles composing excitatory synapses and extrasynaptic membranes of the spines and shafts of pyramidal cells and interneuron dendrites located in the strata oriens, radiatum, and lacunosum-moleculare of the CA1 field of the hippocampus in the two genotypes (wild type and P301S) at 10 months of age. For this purpose, we used the recently developed software GPDQ (Gold Particle Detection and Quantification) to perform automated and semi-automated detection of gold particles present in each compartment of neurons [71]. Most of the spines in the CA1 field arise from pyramidal cells; thus, we refer to them as pyramidal cell spines. Non-spiny dendritic shafts receiving several synapses are considered to originate from interneurons.

Quantitative analysis of the immunogold labeling for GluA1-4 was performed on excitatory postsynaptic specializations and at extrasynaptic sites, indicated by the presence of intramembrane particle (IMP) clusters on the exoplasmic face (E-face) [72]. Excitatory postsynaptic specializations were considered as such when IMP clusters contained at least 30 intramembrane particles [73]; the rest of the neuronal compartment with isolated IMPs was considered as extrasynaptic membrane area. One of the advantages of the SDS-FRL technique is that the whole synaptic specializations of excitatory synapses and extrasynaptic plasma membrane are immediately visible over the surface of neurons. The outline of postsynaptic specialization (IMP clusters) was manually demarcated by connecting the outermost IMP particles, and then we demarcated the rest of the neuronal compartment on the E-face corresponding to the extrasynaptic plasma membrane. The area of synaptic and extrasynaptic sites was measured using the software GPDQ.

Immunogold particles for GluA1-4 were regarded as synaptic labeling if they were within demarcated IMP clusters and those located in the immediate vicinity within 30 nm from the edge of the IMP clusters, given the potential distance between the immunogold particles and antigens. Immunogold particles not meeting these requirements or present in the vicinity of isolated IMPS were regarded as extrasynaptic labeling. Only postsynaptic membrane specializations double labeled for both GluA1-4 and GluN1 were included in the analysis. The number of immunogold particles was counted in both complete and incomplete (partially fractured) postsynaptic membrane specialization. Because the densities of immunogold labeling for the GluA1-4 antibody obtained from complete and incomplete synapses were not significantly different, they were pooled. The density of the immunoparticles for GluA1-4 in each synaptic site was calculated by dividing the number of the immunoparticles by the area of the demarcated IMP clusters. The density of GluA1-4 immunoparticles in the extrasynaptic sites was calculated by dividing the number of the immunoparticles by the area of the demarcated compartment without the synaptic specialization. Measurements were performed in three animals, and results were pooled because the densities for immunogold particles were not significantly different in the different animals. Immunoparticle densities were presented as mean ± SEM between animals.

### 4.7. Controls

To test the method specificity in the procedures for histoblots, the primary antibodies were either omitted or replaced with 5% (*v*/*v*) normal serum of the species of the primary antibody, resulting in total loss of the signal. To test the method specificity in the procedures for electron microscopy, replicas were incubated according to the protocol described above, with primary antibodies omitted or replaced with 1% (*v*/*v*) normal goat serum. The labeling densities on the clusters of intramembrane particles were <1.2 particles/µm^2^ in these cases.

### 4.8. Data Analysis

To avoid observer bias, we performed blinded experiments for immunoblots and immunohistochemistry prior to data analysis. Statistical analyses were performed using GraphPad Prism (San Diego, CA, USA), and data were presented as mean ± SEM unless indicated otherwise. Statistical significance was defined as *p* < 0.05. The statistical evaluation of the immunoblots was performed using the Student *t*-test. To compute the SEM error bars, five blots were measured from each animal. The statistical evaluation of the immunogold densities in the mouse model was performed using the Student *t* test, test for homogeneity of variance, and Shapiro Wilks normality test, and Mann–Whitney *U* test for nonparametric data. Correlations were assessed using Spearman’s correlation test.

## Figures and Tables

**Figure 1 ijms-23-13527-f001:**
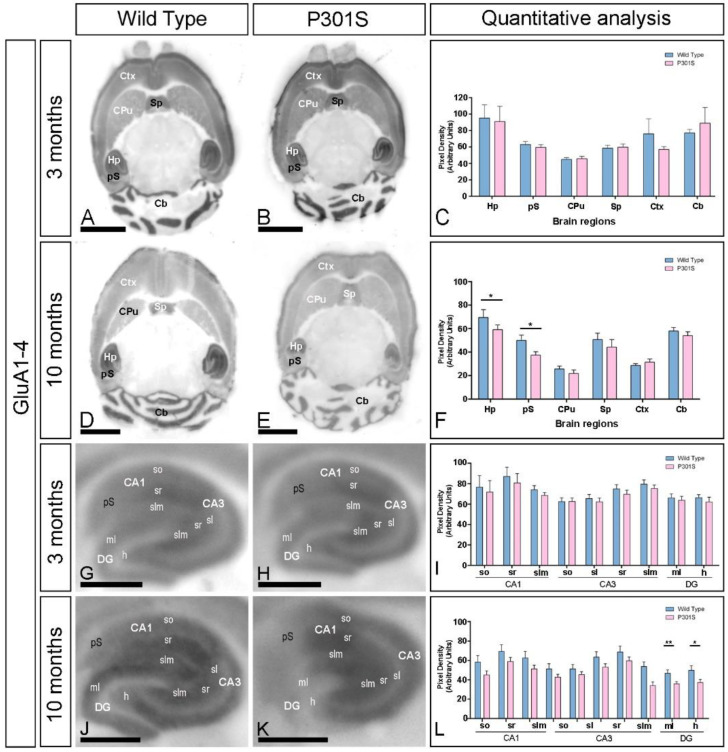
Expression of GluA1-4 in P301S mice. The expression of the GluA1-4 proteins was visualized using histoblots of horizontal brain sections at 3 and 10 months of age in wild-type and P301S mice, using an affinity-purified anti-GluA1-4 antibody. The AMPAR expression in different brain regions was determined by densitometric analysis of the scanned histoblots. (**A**–**F**) The highest level of AMPAR expression was detected in the hippocampus (Hp), followed by the cerebellum (Cb), cortex (Cx), presubiculum (pS) of the hippocampal formation, and septum (Sp). Lower expression levels were detected in the caudate putamen (CPu). Densitometric analysis performed at 3 months showed no differences in the GluA1-4 expression in P301S mice compared to age-matched wild-type controls, but a significant reduction was detected in the hippocampus and presubiculum at 10 months (n = 3 animals per genotype and per age; unpaired *t*-test, * *p* < 0.05). Error bars indicate SEM. Scale bars in (**A**–**E**): 0.25 cm. (**G**–**L**) The expression of GluA1-4 was strong in all dendritic layers of the CA1 and CA3 region and dentate gyrus, with the stratum radiatum (sr) of CA1 and CA3 showing the highest expression level. Densitometric analysis performed at 3 months of age showed no differences in the GluA1-4 expression in P301S mice. At 10 months of age, however, the expression of GluA1-4 was significantly reduced in the molecular layer and hilus of the dentate gyrus of P301S mice (unpaired *t*-test, ** *p* < 0.01, * *p* < 0.05). Error bars indicate SEM. Abbreviations: CA1 region of the hippocampus; CA3, CA3 region of the hippocampus; DG, dentate gyrus; so, stratum oriens; sr, stratum radiatum; slm, stratum lacunosum-moleculare; ml, molecular layer; h, hilus. Scale bars in (**G**–**K**): 0.87 mm.

**Figure 2 ijms-23-13527-f002:**
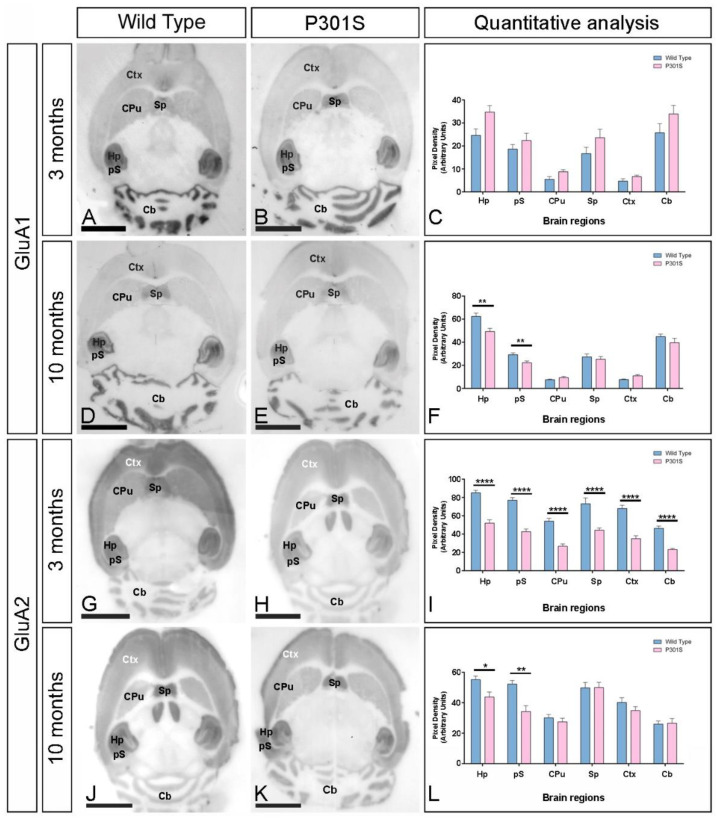
Brain expression of GluA1 and GluA2 in P301S mice. (**A**–**L**) The expression of the GluA1 and GluA2 proteins was visualized using histoblots of horizontal brain sections at 3 and 10 months of age in wild-type and P301S mice, using subunit-specific affinity-purified anti-GluA1 and anti-GluA2 antibodies. The GluA1 expression in different brain regions was determined by densitometric analysis of the scanned histoblots (panels (**C**) and (**F**)). The highest level of GluA1 expression was detected in the hippocampus (Hp), followed by the cerebellum (Cb), the presubiculum (pS) of the hippocampal formation, and the septum (Sp). Lower expression levels were detected in the caudate putamen (CPu) and cortex (Ctx). Densitometric analysis showed a significant reduction in the hippocampus and presubiculum at 10 months in P301S mice compared to age-matched wild-type controls (unpaired *t*-test, ** *p* < 0.01). For the GluA2 subunit, we found significant reductions in all brain regions analyzed at 3 months (n = 3 animals per genotype; unpaired *t*-test, **** *p* < 0.0001) but only a significant reduction in the hippocampus and presubiculum at 10 months in P301S mice compared to age-matched wild-type controls (n = 3 animals per genotype; unpaired *t*-test, * *p* < 0.05, ** *p* < 0.01). Error bars indicate SEM. Scale bars: 0.25 cm.

**Figure 3 ijms-23-13527-f003:**
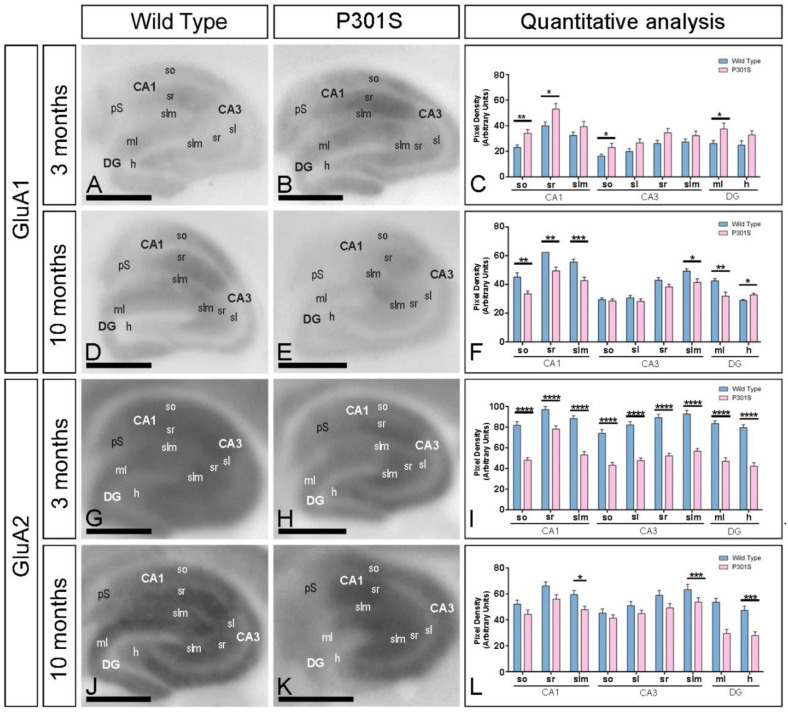
Hippocampal expression of GluA1 and GluA2 in P301S mice. (**A**–**L**) The expression of the GluA1 and GluA2 proteins was visualized using histoblots of horizontal brain sections at 3 and 10 months of age in wild-type and P301S mice, using subunit-specific affinity-purified anti-GluA1 and anti-GluA2 antibodies. The expression of GluA1 and GluA2 was strong in all dendritic layers of the CA1 and CA3 region and dentate gyrus. Densitometric analysis showed a significant increase in the GluA1 expression in SO and SR of CA1, SO of CA3, and ml of the dentate gyrus at 3 months (unpaired *t*-test, ** *p* < 0.01, * *p* < 0.05) but a significant reduction through CA1, SLM of CA3, and ml and hilus of the dentate gyrus at 10 months in P301S mice compared to age-matched wild-type controls (unpaired *t*-test, * *p* < 0.05, ** *p* < 0.01, *** *p* < 0.001, **** *p* < 0.0001). For the GluA2 subunit, we found significant reductions in all hippocampal fields and dendritic layers analyzed at 3 months (n = 3 animals per genotype; unpaired *t*-test, *p* < 0.0001), but only a significant reduction in SLM of CA1 and CA3 and hilus of the dentate gyrus at 10 months in P301S mice compared to age-matched wild-type controls (n = 3 animals per genotype; unpaired t-test, * *p* < 0.05, *** *p* < 0.001). Error bars indicate SEM. Abbreviations: CA1 region of the hippocampus; CA3, CA3 region of the hippocampus; DG, dentate gyrus; so, stratum oriens; sr, stratum radiatum; slm, stratum lacunosum-moleculare; ml, molecular layer; h, hilus. Scale bars: 0.87 mm.

**Figure 4 ijms-23-13527-f004:**
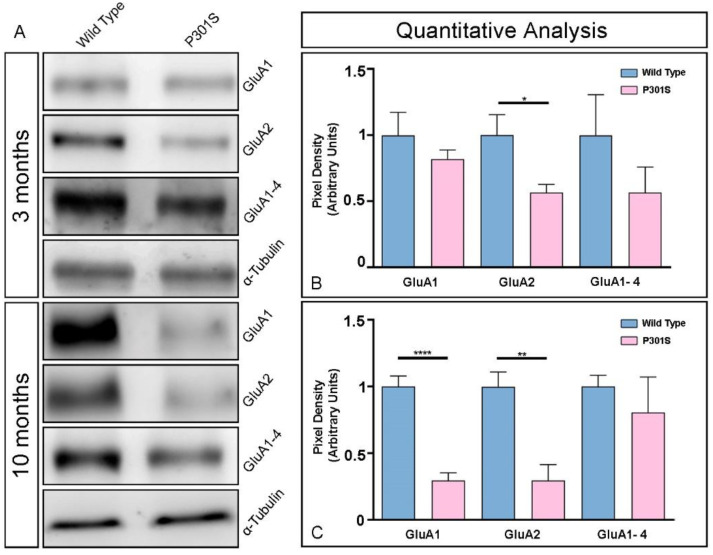
Differential expression of GluA1-4 in the hippocampus between control and P301S mice detected with immunoblots. Representative immunoblots of GluA1, GluA2, and GluA1-4 in the hippocampus from control and P301S mice at 3 and 10 months of age. (**A**) Crude membrane preparations were subjected to 7.5% SDS-PAGE and transferred onto polyvinylidene difluoride membranes. They were reacted with anti-GluA1, anti-GluA2, and anti-GluA1-4 antibodies, which detected a single predominant band with an estimated molecular mass of 100 kDa. (**B**,**C**) The developed immunoblots were scanned and densitometric measurements from five independent experiments were averaged together to compare the protein densities for each age (3 months in panel (**B**) and 10 months in panel (**C**)) and experimental group (n = 3 animals per genotype and per age). The developed immunoblots were scanned and densitometric measurements were averaged together to compare the protein densities between wild-type and P301S mice in the hippocampus. Quantification of GluA1, GluA2, and GluA1-4 was normalized to α-tubulin and expressed as the pixel density. Data showed a significant reduction in the amount of GluA2 protein at 3 months, and GluA1 and GluA2 at 10 months in P301S mice (Mann–Whitney test, * *p* < 0.01, ** *p* < 0.001, **** *p* < 0.00001). Data are means ± SEM.

**Figure 5 ijms-23-13527-f005:**
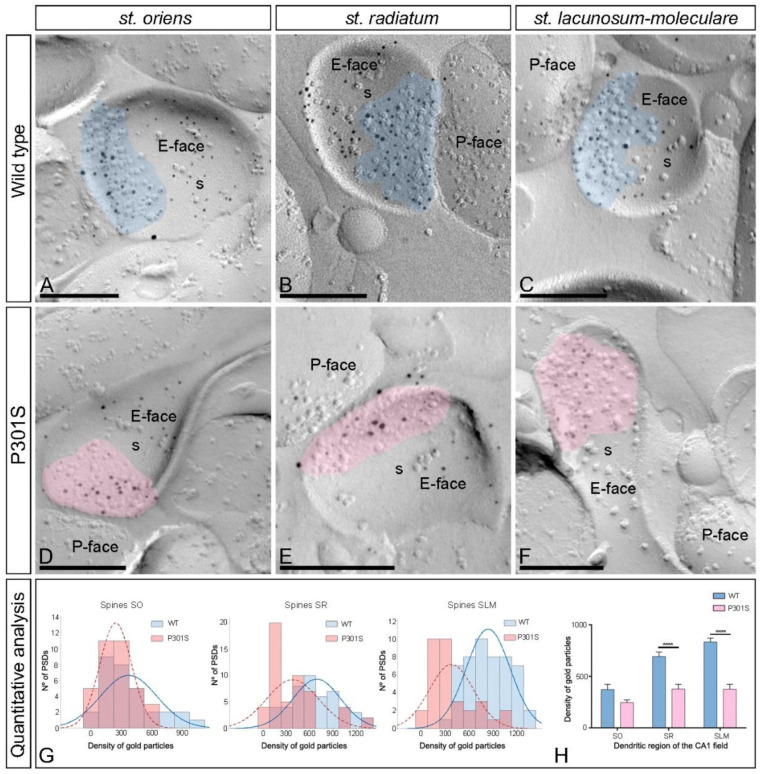
Alteration of synaptic GluA1-4 in the spines of P301S mice. (**A**–**F**) Electron micrographs of SDS-FRL replicas simultaneously double immunolabeled with antibodies against the GluA1-4 subunits of AMPARs (5 nm gold particles) and the GluN1 subunit of NMDARs (10 nm gold particles) in the hippocampus of P301S mice at 10 months of age. Postsynaptic membrane specializations (IMP clusters, pseudo colored with transparency in blue for wild type and in red for P301S) show immunoreactivity for both GluA1-4 and GluN1 at the excitatory synaptic sites of the dendritic spines of pyramidal cells in the strata oriens, radiatum, and lacunosum-moleculare in wild-type and P301S mice. (**G**) Histograms showing the distribution of densities of immunoparticles for GluA1-4 of individual postsynaptic membrane specializations in the hippocampal CA1 field in wild-type and P301S mice. (**H**) Quantitative analysis showing the mean densities of GluA1-4 in the excitatory synapses in the spines. A significant reduction in the density of GluA1-4 immunoparticles was detected in the spines located in the strata radiatum and lacunosum-moleculare, but no changes were found in the stratum oriens of the CA1 field of P301S mice compared to age-matched wild-type (unpaired *t*-test, **** *p* < 0.0001). Scale bars: (**A**,**C**–**F**), 200 nm; (**B**), 250 nm.

**Figure 6 ijms-23-13527-f006:**
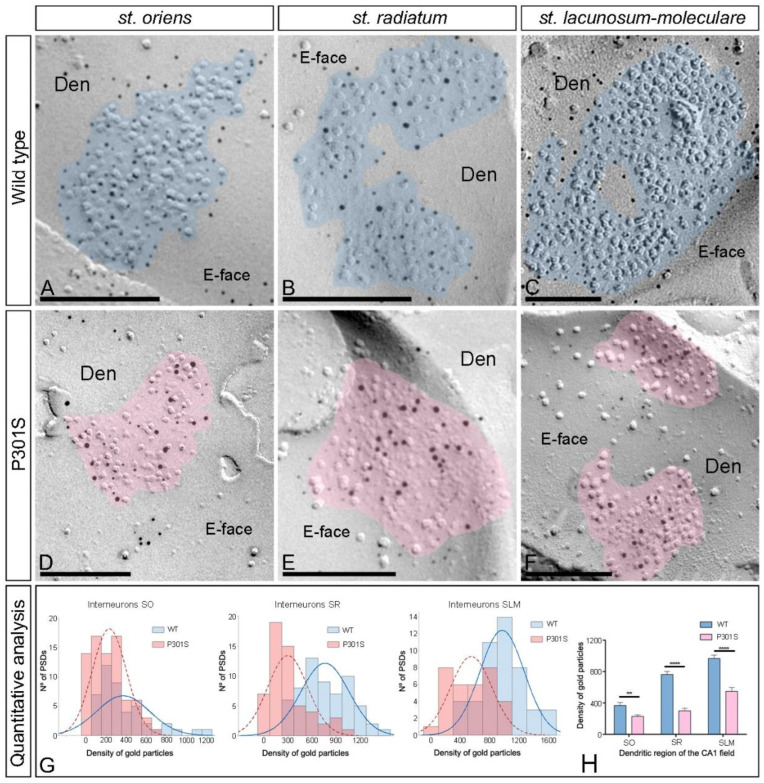
Alteration of synaptic GluA1-4 in the interneurons of P301S mice. (**A**–**F**) Electron micrographs of SDS-FRL replicas simultaneously double immunolabeled with antibodies against the GluA1-4 subunits of AMPARs (5 nm gold particles) and the GluN1 subunit of NMDARs (10 nm gold particles) in the hippocampus of P301S mice at 10 months of age. Postsynaptic membrane specializations (IMP clusters, pseudo colored with transparency in blue for wild type and in red for P301S) show immunoreactivity for both GluA1-4 and GluN1 at the excitatory synaptic sites of the interneuron dendrites in the strata oriens, radiatum, and lacunosum-moleculare in wild-type and P301S mice. (**G**) Histograms showing the distribution of densities of immunoparticles for GluA1-4 of individual postsynaptic membrane specializations in the hippocampal CA1 field in wild-type and P301S mice. (**H**) Quantitative analysis showing the mean densities of GluA1-4 in the excitatory synapses in the spines. A significant reduction in the density of GluA1-4 immunoparticles was detected in the spines located in the three strata of the CA1 field of P301S mice compared to age-matched wild-type mice (unpaired *t*-test, ** *p* < 0.01, **** *p* < 0.0001). Scale bars: (**A**,**C**), 250 nm; (**B**,**D**–**F**), 200 nm.

**Figure 7 ijms-23-13527-f007:**
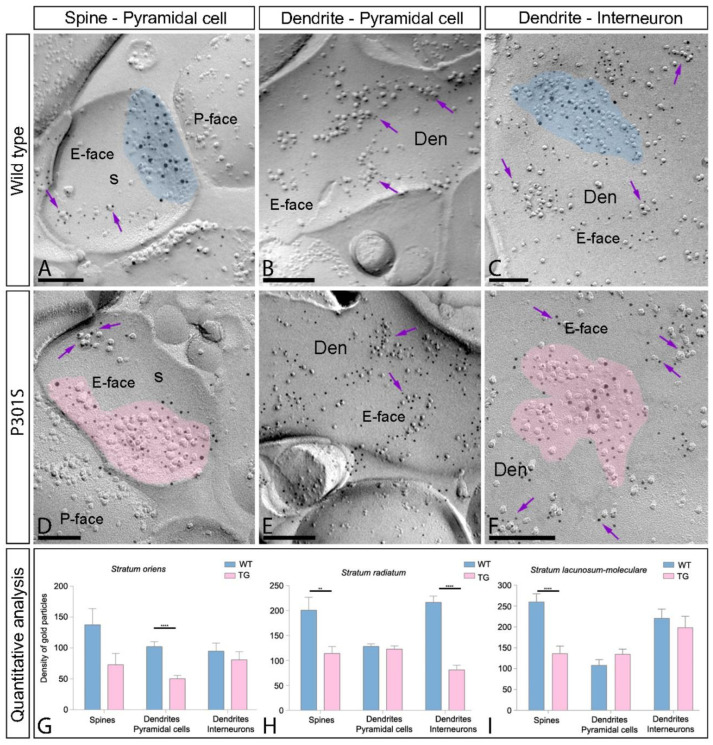
Reduced density of extrasynaptic GluA1-4 in the hippocampus of P301S mice. (**A**–**F**) Electron micrographs of the hippocampus showing double labeling for the GluA1-4 subunits of AMPARs (5 nm gold particles) and the GluN1 subunit of NMDARs (10 nm gold particles) at the extrasynaptic sites of spines and dendrites of the pyramidal cells and interneuron dendrites in the CA1 field, as detected using the SDS-FRL technique in P301S mice at 10 months of age. For illustration purposes, the electron micrographs belong to the stratum radiatum. Immunoparticles for GluA1-4 (5 nm gold particles, purple arrows) were associated with IMPs outside postsynaptic membrane specializations (pseudo colored with transparency in blue for wild type and in pink for P301S) considered as extrasynaptic membrane compartments. (**G**–**I**) Quantitative analysis showing the mean densities of GluA1-4 in the extrasynaptic compartments in the three dendritic layers of the CA1 field. A significant decrease in the density of GluA1-4 immunoparticles at the extrasynaptic sites was detected in the spines of the strata radiatum and lacunosum-moleculare, in the dendritic shafts of pyramidal cells in the stratum oriens, and in interneuron dendrites distributed in the stratum radiatum of P301S mice compared to age-matched wild-type mice (unpaired *t*-test, ** *p* < 0.01, **** *p* < 0.0001). No differences were detected in all other compartments throughout the CA1 field (unpaired *t*-test, *p* > 0.1). Scale bars: (**A**–**F**), 200 nm.

**Table 1 ijms-23-13527-t001:** Number and density of gold particles for pan-AMPA at different excitatory synapses in the CA1 region at 10 months of age. Density values are provided in immunogold/μm^2^.

	*Stratum oriens*		*Stratum radiatum*		*Stratum lacunosum-moleculare*	
	Spines	Interneurons	Spines	Interneurons	Spines	Interneurons
**WT**						
Excitatory Synapses (n)	31	45	48	61	51	47
Area of synapses (µm)	0.036 ± 0.003	0.061 ± 0.004	0.032 ± 0.002	0.055 ± 0.004	0.033 ± 0.002	0.063 ± 0.005
Median gold particles	9	19	18	29	22	50
Range	53–1	87–2	76–4	162–9	102–8	199–11
Density gold particles (µm^2^)						
Mean (±SEM)	373.25 ± 49.93	367.0 ± 39.31	692.70 ± 44.43	764.07 ± 38.37	833.86 ± 38.45	967.70 ± 43.95
Median	333.87	301.12	627.02	723.18	818.89	1003.70
Range	1065.06–30.72	1172.68–52.22	1314.80–185.87	1455.36–292.7	1276.96–294.75	1665.71–378.67
**P301S**						
Excitatory Synapses (n)	35	78	52	56	33	33
Area of synapses (µm)	0.034 ± 0.004	0.069 ± 0.004	0.029 ± 0.002	0.067 ± 0.006	0.031 ± 0.002	0.068 ± 0.007
Median gold particles	6	11	7.5	10	8	28
Range	47–1	62–1	47–1	151–2	43–1	135–3
Density gold particles (µm^2^)						
Mean (±SEM)	245.44 ± 26.64	230.02 ± 19.28	377.41 ± 46.73	300.01 ± 33.29	375.59 ± 47.96	548.08 ± 49.05
Median	253.43	224.24	261.39	227.44	316.55	535.08
Range	575.28–26.03	695.81–14.84	1390.82–26.75	981.99–34.91	1033.89–53.82	1001.05–66.32

**Table 2 ijms-23-13527-t002:** Number and density of gold particles for pan-ampa at extrasynaptic compartments in the CA1 region at 10 months of age. Density values are provided in immunogold/μm^2^.

	*Stratum oriens*			*Stratum radiatum*			*Stratum lacunosum-moleculare*		
	PC Spines	PC Dendrites	Interneurons	PC Spines	PC Dendrites	Interneurons	PC Spines	PC Dendrites	Interneurons
**WT**									
Extrasynaptic compartments (n)	23	25	38	38	30	58	50	30	48
Density gold particles (µm^2^)									
Mean (±SEM)	137.83 ± 26.12	102.29 ± 7.75	94.94 ± 13.07	201 ± 26.02	128.3 ± 5.41	216.7 ± 12.47	260.3 ± 19.13	108.7 ± 12.89	221.1 ± 21.58
Median	91	102	71.56	149.3	122.5	207.5	267.6	85.46	186.2
Range	401.3–16.94	187.8–3.95	306.3–5.95	711.3–19.26	196.1–89.92	603.2–74.33	629.2–25.67	332–14.98	765.4–11.17
**P301S**									
Extrasynaptic compartments (n)	15	31	42	39	23	40	26	28	28
Density gold particles (µm^2^)									
Mean (±SEM)	72.86 ± 18.23	50.29 ± 5.71	81.10 ± 13.02	114 ± 13.97	126.1 ± 6.76	81.28 ± 9.40	136.8 ± 17.28	135.2 ± 11.64	198.8 ± 26.52
Median	40.97	42.31	58.28	92.55	117.2	60.78	140.3	150.5	197.6
Range	234.5–18.72	126.7–5.55	406.4–2.25	442.1–11	191.3–70.83	243.1–6.69	382.4–20.68	291.3–15.00	612.7–19.97

## Data Availability

All data used and/or analyzed during the current study are available from the corresponding author on reasonable request.

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
