# Peer review of "Alteration in the Synaptic and Extrasynaptic Organization of AMPA Receptors in the Hippocampus of P301S Tau Transgenic Mice"

_ijms, 2022, doi:10.3390/ijms232113527_

Round 1

Reviewer 1 Report

The study by Alfaro-Ruiz et al. nicely shows a reduction in some AMPA subunits and changes in their synaptic distribution. Synaptic dysfunction is an important hallmark of neurodegenerative diseases, including Alzheimer's. Multiple attempts exist to develop biomarkers for synaptic dysfunction, including CSF biomarkers and SV2 PET. The current work shows that AMPAR level is primarily unchanged in P301S mice at three months of age and shows a minor but significant reduction in some brain regions at 10 months of age. However, they also show that the composition of AMPAR is significantly changing with a decrease of GluR2 and an increase in GluR1, They also nicely show early changes in the distribution of the receptor across the synapse.  

These observations can help develop new biomarkers, for example, based on their data, GluR2 seems to be the best candidate and suggest how tau pathology effect AMPAR. 

However, my main critique is that this study is only observational and relay on just one experimental model.  Therefore, while I found the methods to be solid and the data to be well presented, the conclusions are overreaching, in my opinion. This study does not provide mechanistic data for tau pathology or even conclusive evidence that the observed changes in AMPAR play a central role in tau pathology.  

I understand that providing such evidence requires a significant amount of resources and work and can significantly delay the publication of these interesting observations and thus suggest a major edit of the conclusion to reflect the current findings. Below are some examples of statements I found to be overreaching, but I recommend that authors carefully review and edit the discussion.     

* In line 402-404 " layers, we provide compelling evidence implicating the role of GluA1/2 receptor subunit-specific trafficking in tau pathology driving to the decline of AMPARs at excitatory synapses that could culminate in synapse and 404 spine loss in P301S mice [21]."  I think that to support such a claim, the changes in GluA1/2 should be inhibited without changing tau aggregation and then show that the mice live longer or have fewer symptoms. 

* Line 478-480 "Here, our data strongly provide clear evidence suggesting that tau induces synaptic dysfunction through the decline of surface AMPARs." same as above

* line 482 " In summary, the present work provides evidence that hippocampal GluA1 and GluA2 subunits play an essential role in tau pathology." same as above

* One more claim in the discussion that I found inaccurate is in lines 351-353 "Increasing evidence indicates that loss of synapses correlates better with tau pathology than with amyloid pathology and that tau neurotoxicity is downstream of amyloidosis [27–30]." There is evidence that Tau pathology better correlates with the pathology but not necessarily with synaptic dysfunction. The authors of this paper previously reported changes in AMPAR synaptic distribution in APP/PS1 mice that do not have tau pathology. Moreover, while I do not think current data provide clear evidence, most biomarker studies suggest that Amyloid pathology starts before Tau and is not downstream of it. 

Author Response

Reviewer 1:

We are grateful for the valuable suggestions of the Referee that we have incorporated into the manuscript. The changed words have been highlighted in bold in the manuscript.

Q 1 - Specific comment: “The study by Alfaro-Ruiz et al. nicely shows a reduction in some AMPA subunits and changes in their synaptic distribution. Synaptic dysfunction is an important hallmark of neurodegenerative diseases, including Alzheimer's. Multiple attempts exist to develop biomarkers for synaptic dysfunction, including CSF biomarkers and SV2 PET. The current work shows that AMPAR level is primarily unchanged in P301S mice at three months of age and shows a minor but significant reduction in some brain regions at 10 months of age. However, they also show that the composition of AMPAR is significantly changing with a decrease of GluR2 and an increase in GluR1. They also nicely show early changes in the distribution of the receptor across the synapse.  // These observations can help develop new biomarkers, for example, based on their data, GluR2 seems to be the best candidate and suggest how tau pathology effect AMPAR. // However, my main critique is that this study is only observational and relay on just one experimental model.  Therefore, while I found the methods to be solid and the data to be well presented, the conclusions are overreaching, in my opinion. This study does not provide mechanistic data for tau pathology or even conclusive evidence that the observed changes in AMPAR play a central role in tau pathology. I understand that providing such evidence requires a significant amount of resources and work and can significantly delay the publication of these interesting observations and thus suggest a major edit of the conclusion to reflect the current findings. Below are some examples of statements I found to be overreaching, but I recommend that authors carefully review and edit the discussion”.

Authors’ response: We acknowledge the Reviewer for these comments. Following his/her suggestion, we have carefully reviewed the discussion. All edits, including those suggested bellow, have been highlighted in bold in the manuscript.

Q 2 - Specific comment:In line 402-404 " layers, we provide compelling evidence implicating the role of GluA1/2 receptor subunit-specific trafficking in tau pathology driving to the decline of AMPARs at excitatory synapses that could culminate in synapse and spine loss in P301S mice [21]."  I think that to support such a claim, the changes in GluA1/2 should be inhibited without changing tau aggregation and then show that the mice live longer or have fewer symptoms.”.

Authors’ response: We acknowledge the Reviewer for this comment. We have re-written the sentence, changing “we provide compelling evidence implicating” by “our data favors a possible role” in lines 398-401. The sentence now reads as follow: “…layers, our data favors a possible role of GluA1/2 receptor subunit-specific trafficking in tau pathology driving to the decline of AMPARs at excitatory synapses that could culminate in synapse and spine loss in P301S mice [21]”.

Q 3 - Specific comment:Line 478-480 "Here, our data strongly provide clear evidence suggesting that tau induces synaptic dysfunction through the decline of surface AMPARs." same as above”.

Authors’ response: We acknowledge the Reviewer for this comment. We have re-written the sentence in lines 474-476. The sentence now reads as follow: “Here, our quantitative ultrastructural data support de view that tau might induce synaptic dysfunction [36] through the decline of surface AMPARs”.

Q 4 - Specific comment:line 482 " In summary, the present work provides evidence that hippocampal GluA1 and GluA2 subunits play an essential role in tau pathology." same as above”.

Authors’ response: We acknowledge the Reviewer for this comment. We have re-written the sentence in lines 477-478. The sentence now reads as follow: “In summary, the present work provides evidence that hippocampal GluA1 and GluA2 subunits are altered in tau pathology”.

Q 5 - Specific comment:One more claim in the discussion that I found inaccurate is in lines 351-353 "Increasing evidence indicates that loss of synapses correlates better with tau pathology than with amyloid pathology and that tau neurotoxicity is downstream of amyloidosis [27–30]." There is evidence that Tau pathology better correlates with the pathology but not necessarily with synaptic dysfunction. The authors of this paper previously reported changes in AMPAR synaptic distribution in APP/PS1 mice that do not have tau pathology. Moreover, while I do not think current data provide clear evidence, most biomarker studies suggest that Amyloid pathology starts before Tau and is not downstream of it.”.

Authors’ response: We acknowledge the Reviewer for this comment. We have changed the sentence in lines 447-448 by deleting “and that tau neurotoxicity is downstream of amyloidosis”. The sentence now reads as follow: “Increasing evidence indicates that loss of synapses correlates better with tau pathology than with amyloid pathology [27–30]”.

Reviewer 2 Report

Alteration in the synaptic and extrasynaptic organization of AMPA receptors in the hippocampus of P301S tau transgenic mice by Alfaro-Ruiz er al. The authors present a clear and interesting paper regarding the expressing of AMPA receptors throughout the brain with focus on the Hippocampus area. Overall this paper is well written and I only have one minor comments to Figure 1: In the left side of Figure 1 add the label: GluA1-4 (same style as Figure 2 & 3).

Author Response

Reviewer 2:

We are grateful for the valuable suggestions of the Referee that we have incorporated into the manuscript.

Q 1 - Specific comment:Alteration in the synaptic and extrasynaptic organization of AMPA receptors in the hippocampus of P301S tau transgenic mice by Alfaro-Ruiz er al. The authors present a clear and interesting paper regarding the expressing of AMPA receptors throughout the brain with focus on the Hippocampus area. Overall this paper is well written and I only have one minor comments to Figure 1: In the left side of Figure 1 add the label: GluA1-4 (same style as Figure 2 & 3)”.

Authors’ response: We acknowledge the Reviewer for this useful observation. Following the suggestion of the reviewer, we have modified Figure 1 in the resubmitted version of this manuscript.